# Significance of NT-proBNP and High-Sensitivity Troponin in Friedreich Ataxia

**DOI:** 10.3390/jcm9061630

**Published:** 2020-05-28

**Authors:** Lise Legrand, Carole Maupain, Marie-Lorraine Monin, Claire Ewenczyk, Richard Isnard, Rana Alkouri, Alexandra Durr, Francoise Pousset

**Affiliations:** 1Cardiology Department, AP-HP, Sorbonne Université, Pitié-Salpêtrière University Hospital, 75013 Paris, France; lise.legrand@aphp.fr (L.L.); carole.maupain@aphp.fr (C.M.); richard.isnard@aphp.fr (R.I.); 2ICAN (Institute for Cardiometabolism and Nutrition), Pitié-Salpêtrière University Hospital, 75651 Paris, France; 3ACTION (Allies in Cardiovascular Trials Initiatives and Organized Networks) Group, URC Lariboisière University Hospital, 75475 Paris, France; 4Institut du Cerveau et de la Moelle épinière (ICM), AP-HP, INSERM, CNRS, Sorbonne Université, Pitié-Salpêtrière University Hospital, 75013 Paris, France; marie-lorraine.monin@aphp.fr (M.-L.M.); claire.ewenczyk@aphp.fr (C.E.); alexandra.durr@icm-institute.org (A.D.); 5Metabolic Biochemistry Department, Sorbonne Université, Pitié-Salpêtrière University Hospital, 75013 Paris, France; rana.alkouri@aphp.fr

**Keywords:** Friedreich ataxia, hs troponin, NT-proBNP, cardiomyopathy

## Abstract

Background: Friedreich’s ataxia (FA) is a rare autosomal recessive mitochondrial disease resulting of a triplet repeat expansion guanine-adenine-adenine (GAA) in the frataxin (FXN) gene, exhibiting progressive cerebellar ataxia, diabetes and cardiomyopathy. We aimed to determine the relationship between cardiac biomarkers, serum N-terminal pro-brain natriuretic peptide (NT-proBNP), and serum cardiac high-sensitivity troponin (hsTnT) concentrations, and the extent of genetic abnormality and cardiac parameters. Methods: Between 2013 and 2015, 85 consecutive genetically confirmed FA adult patients were prospectively evaluated by measuring plasma hsTnT and NT-proBNP concentrations, electrocardiogram, and echocardiography. Results: The 85 FA patients (49% women) with a mean age of 39 ± 12 years, a mean disease onset of 17 ± 11 years had a mean SARA (Scale for the Assessment and Rating of Ataxia) score of 26 ± 10. The median hsTnT concentration was 10 ng/L (3 to 85 ng/L) and 34% had a significant elevated hsTnT ≥ 14 ng/L. Increased septal wall thickness was associated with increased hsTnT plasma levels (*p* < 0.001). The median NT-proBNP concentration was 31 ng/L (5 to 775 ng/L) and 14% had significant elevated NT-proBNP ≥ 125 ng/L. Markers of increased left ventricular filling pressure (trans mitral E/A and lateral E/E’ ratio) were associated with increased NT-proBNP plasma levels (*p* = 0.01 and *p* = 0.01). Length of GAA or the SARA score were not associated with hsTnT or NT-proBNP plasma levels. Conclusion: hsTnT was increased in 1/3 of the adult FA and associated with increased septal wall thickness. Increased NT-proBNP remained a marker of increased left ventricular filling pressure. This could be used to identify patients that should undergo a closer cardiac surveillance.

## 1. Introduction

Friedreich ataxia (FA) is an autosomal recessive mitochondrial disorder and the most frequent inherited ataxia in Europe. It is characterized by progressive spinocerebellar ataxia and extra neurological signs, such as scoliosis, diabetes, and cardiomyopathy. First symptoms often occur during adolescence or occasionally later in life. The age of onset has a profound effect on the disease severity with a faster deterioration associated with a younger age of onset. FA is in most cases caused by a homozygous guanine-adenine-adenine (GAA) trinucleotide expansion in the frataxin (FXN) gene, which encodes frataxin, a mitochondrial protein involved in iron metabolism [1,2,3]. The level of functional frataxin expressed depends on the number of GAA repeats in the shorter allele of the FXN gene (GAA1). Longer GAA1 are associated with earlier symptom onset and increased severity of the neurologic disease [3,4]. The extent of the association of GAA1 with the cardiopathy was variable between studies [5,6], the number of GAA repeat in the larger allele (GAA2) appeared to be less closely related than GAA1 to clinical variables [6].

The neurological syndrome is well characterized but our understanding of the nature of cardiac involvement is still incomplete. A concentric pattern of left ventricular geometry appears to be common and hypertrophic cardiomyopathy is observed in two thirds of FA patients [1,6]. Cardiac disease is the major cause of premature mortality in these patients, mostly due to heart failure and supraventricular arrhythmia [6,7]. Independent predictive factors of survival in Friedreich ataxia appear to be length of GAA1 and cardiac parameters, left ventricular mass and left ventricular ejection fraction [6].

Cardiac biomarkers, such as plasma levels of cardiac troponin and N-terminal pro-brain natriuretic peptide (NT-proBNP), are easily available and widely used in cardiology. Troponin is a marker of cardiomyocyte injury and has emerged as the preferred biomarker for the diagnosis of acute myocardial infarction. Cardiac troponin has also a prognostic value in chronic heart failure with plasma levels generally lower than in coronary disease [8]. Elevated plasma troponin levels have been reported previously in FA patients with different significance according to the studies: common baseline characteristics or markers of advanced stage of the cardiopathy [5,9].

NT-proBNP is released by the heart in response to myocardial stretch and increased intravascular volume: it is considered to be a surrogate marker of elevated left ventricular filling pressure. The plasma level of NT-proBNP is a diagnostic and prognostic marker in patients with and without heart failure [10,11,12]. However, values of natriuretic peptides in FA patients are still unknown.

These two cardiac biomarkers could be useful for the evaluation of cardiac involvement in FA patients. The objective of this study was to determine cardiac troponin and NT-proBNP plasma levels in a large cohort of FA adult patients, and to investigate whether relationships exist with the extent of the genetic abnormality and/or with conventional echocardiographic markers of cardiac involvement.

## 2. Methods

### 2.1. Population

This study was an observational cross-sectional cohort study; data were collected prospectively. The cohort consisted of consecutively patients with genetically confirmed FA evaluated in the Department of Cardiology (Pitié-Salpêtrière University Hospital, Paris, France) between 1st July 2013 and 31st December 2015. Patients were aged > 16 years at evaluation. The Institut National de la Santé et de la Recherche Médicale (INSERM) approved the study. Informed written consent was given in accordance with the law and local ethical regulations. Forty-seven patients were included as part of a prospective European EFACTS registry (European Friedreich Ataxia Consortium for Translational Studies) [3,4].

### 2.2. Neurological Evaluation

Neurological assessment was performed by a neurologist within 12 months of cardiac evaluation, and was assessed using the SARA score (Scale for the Assessment and Rating of Ataxia), ranging from 0 to 40. It is a semi quantitative scale developed to assess ataxia with values from 0 (no ataxia) to 40 (most severe ataxia) [13].

Cardiac evaluation—including clinical evaluation, standard 12-lead electrocardiography, and standard transthoracic echocardiography—was performed on the same day as blood samples were collected.

### 2.3. Electrocardiogram (ECG)

A 12-lead surface resting ECG was recorded at 25 mm/s and 10 mm/mV and analyzed by one experienced cardiologist (L.L.) blinded from the echocardiographic and biological results. Repolarization was considered abnormal in the presence of negative T waves (≥0.1 mV) or flat T waves in at least two leads, except V1 and V2 in the absence of bundle branch block.

### 2.4. Echocardiography

Echocardiography was performed using a Vivid-E9 (General Electric Company (GE Healthcare, Horten, Norway)) by the same expert cardiologist (F.P.) blinded to biological and ECG data. Each measure was the mean of three measurements. Left ventricular end diastolic diameter (LVDD), left ventricular end systolic diameter (LVSD), septal wall thickness (SWT), and posterior wall thickness (PWT) were measured in the parasternal long axis views using standard M mode according to the recommendations of the American Society of Echocardiography [14]. Relative wall thickness was defined by the formula: RWT = (SWT + PWT)/LVDD(1)

Pathological concentric LV remodeling was defined as an RWT exceeding 0.42 [15]. Left ventricular mass (LVM) was calculated using the formula:LVM = 0.8 × (1.04 (LVDD + PWT + SWT)^3^ − LVTD^3^) + 0.6 g(2)
which was indexed to the body surface area to obtain the LV mass index (LVMi). Left ventricular ejection fraction was calculated using the Simpson’s biplane method based on 2- and 4-chamber views [14] Peak E and A wave were measured by vascular Doppler on the mitral valve. Tissue Doppler imaging was performed on the lateral wall of the left ventricle (E’). An estimate of LV filling pressure was made using the ratio between trans mitral peak E velocity and the peak E’ velocity measured at the lateral wall (E/E’) [16]. Left atrial size was measured by left atrial area in an apical four-chamber view [14]. Cardiac hypertrophy was defined, according to the Henry nomogram, by SWT or PWT ≥ 95% of the calculated value for age and body surface area [17].

### 2.5. Blood Assays

Both serum NT-proBNP and cardiac troponin levels were measured in peripheral venous blood samples on the day of collection. During the study, the method used by the hospital laboratory to measure cardiac troponin changed on the 1st January 2014. For the first 15 patients included in 2013, cardiac troponin was measured by a conventional method, and for the following 70 patients, high sensitivity cardiac troponin T (hsTnT) was measured. Only high sensitivity troponin values were used for further analysis.

Plasma and serum were separated by centrifugation and serum hsTnT and NT-proBNP were measured in plasma by the central biochemistry laboratory of the Pitié-Salpêtrière University Hospital, using Roche Diagnostics^®^ (Roche Diagnostics, Meylan, France) assays and a Roche *COBAS* analyzer in accordance with the manufacturer’s instructions. The results are reported in ng/L with a detection limit of 3 ng/L for hsTnT and 5 ng/L for NT-proBNP. When the measurement value was below the detection limit, the detection limit value was used. For hsTnT assays, the Coefficients of Variation (CVs) of the laboratory on control samples were below or equal to 2.5% for low levels and below or equal 3% for high levels. For NT-proBNP assays, CVs on control samples were below or equal to 3.5% for low levels and below or equal to 5.3% for high levels during the period of the study.

Normal values for hsTnT and NT-proBNP were defined according to the 99th percentile range of the population previously reported and were <14 ng/L and <125 ng/L, respectively.

### 2.6. Statistical Analysis

Descriptive statistics are presented as proportions for categorical variables, means ± SD for continuous variables and median (min, max) for NT-proBNP and cardiac troponin. As the normality assumption of hsTnT and NT-proBNP was not confirmed, we dichotomized these parameters into categorical variables. The thresholds were defined according to the 99th percentile range of the general population and were < or ≥ 14 ng/L for HsTnT and < or ≥ 125 ng/L for NT-proBNP. Septal hypertrophy assessed by septal wall thickness in mm was compared using Mann Whitney U test across the two groups of patients with hsTnT < or ≥ 14 ng/L.

We then performed two different multivariate logistic regression analysis. The first model assessed the relationship between hsTnT and cofactors (clinical, genetics, and echographic parameters). The second model assessed the relationship between NT-proBNP and cofactors (clinical, genetics, and echographic parameters). Variables with *p* values < 0.1 by univariate analysis were included in the multivariate analysis. The multivariate analysis was a forward stepwise analysis with a *p*-value of 0.05 for significance level for addition to the model. The first-order interactions in multivariable analysis were investigated (*p* < 0.1). Confidence intervals were 95%. Analyses were performed using STATA/SE version 13.0, StataCorp LLC, College Station, TX, USA.

## 3. Results

### 3.1. Clinical Characteristics of the Population

A total of 85 FA patients were evaluated in cardiology during the study period with a mean age of 39 ± 12 years (16 to 79 years). Mean GAA1 was 480 ± 300 pb and mean GAA2 was 750 ± 300 pb. Two patients carried heterozygous point mutations in association with GAA repeat (482 + 1G > C and 460A > T).

Clinical characteristics of the patients are presented in Table 1. The mean age of disease onset was 17 ± 11 years; 53/85 (62%) were wheelchair bound, the mean age of onset of being wheelchair bound was 26 ± 10 years. The mean SARA score was 26 ± 10 for 80/85 (94%) patients. There were 9/85 (10.5%) subjects with a diagnosis of diabetes and 7/85 (8%) with a diagnosis of hypertension. No patient had history of coronary disease. Eight of 85 (9%) had previous paroxysmal or persistent atrial fibrillation, and 3/85 (3%) previous heart failure.

Overall, 9/85 (10.5%) patients were on beta-blockers, 9/85 (10.5%) on ACE inhibitors, 7/85 (8%) on oral anticoagulants (vitamin K antagonist and NOAC), 4/85 (5%) on amiodarone, and 1/85 (1%) on flecainide. One patient had a pacemaker for bradyarrythmia,

At inclusion, all patients were normotensive, 2/85 (2%) patients experienced dyspnea, 3/85 (3%) chest pain, and 3/85 (3%) palpitations, but no patients had either clinical indicators of coronary disease or signs of heart failure.

#### 3.1.1. ECG

Cardiac involvement assessed by ECG was present in the majority of the population: 75/85 (88%) patients had abnormal repolarization as described in the methods; 13/85 (15%) had a right bundle branch block (complete or incomplete). Patients were in sinus rhythm, except 3/85 (3%) with persistent atrial fibrillation.

#### 3.1.2. Echocardiography

Echocardiographic parameters are shown in Table 1. We observed left ventricular structural abnormalities, assessed by an RWT > 0.42, in 71% of patients. Patients had small ventricle with a mean LVDD of 45 ± 5 mm. According to Henry’s nomograms, 51/85 (60%) patients had LV hypertrophy. Only 16 patients had SWT or PWT ≥ 13 mm and 3 ≥ 15 mm, and the sex-specific LV mass index was increased in 31% (10 men and 16 women). Left ventricle outflow tract obstruction was not observed.

All but two patients (LVEF of 37% and 46%) had a preserved LV ejection fraction (>50%). Left atria were not dilated (area < 20 cm^2^) and E/A and lateral E/E’ were within the normal range (E/E’ < 8).

#### 3.1.3. Cardiac Biomarkers

hsTnT (high sensitivity cardiac troponin T)**:** As mentioned previously, hsTnT plasma values were not available for the first 15 patients included in the study. For the following 70 patients, the median value for hsTnT was 10 ng/L and ranged from 3 to 85 ng/L. No patient had significant values for the diagnosis of acute coronary syndrome and 26/70 (33%) had elevated hsTnT ≥ 14 ng/L. Length of GAA1 and GAA2 was not associated with elevated plasma hsTnT levels (*p* = ns).

Factors reflected hypertrophy (LV mass index, septal wall thickness, and posterior wall thickness) were associated with elevated plasma hsTnT levels in univariate analysis but not RWT, reflecting remodeling (Table 2). In multivariate analysis, increased septal wall thickness was the only independent factor associated with elevated plasma hsTnT levels (*p* < 0.001). Patients with hsTnT ≥ 14 ng/L had significantly increased LV septal wall thickness (Figure 1). Neither left ventricular ejection fraction nor plasma NT-proBNP levels were associated with elevated plasma hsTnT levels.

NT-proBNP (N-terminal pro-brain natriuretic peptide): The median value for NT-proBNP was 31 ng/L and ranged from 5 to 775 ng/L in 85 patients. Twelve of 85 (14%) patients had elevated NT-proBNP ≥ 125 ng/L. Factors associated with elevated plasma NT-proBNP levels are shown in Table 3. In univariate analysis, several factors are associated with NT-proBNP plasma levels, including GAA1 (*p* = 0.02). In multivariate analysis only E/A ratio (*p* = 0.01) and E/E’ ratio (*p* = 0.01) were independently associated with elevated NT-proBNP concentrations.

In univariate analysis, the SARA score was not significantly associated with NT-proBNP or hsTnT plasma levels.

Elevated hsTnT and NT-proBNP: Six patients (five female) had hsTnT ≥ 14 ng/L and NT-proBNP ≥ 125 ng/L with median plasma hsTnT level of 23.6 ng/L (17.6 to 85.4 ng/L) and median NT-proBNP level of 458 ng/L (131 to 775 ng/L). These patients were aged 35 ± 18 years, with mean GAA1 of 633 ± 200 bp, SWT of 13 ± 2.7 mm, PWT of 10.4 ± 2 mm, and LVEF of 56 ± 14%; In the pas, 3/6 patients had previous atrial fibrillation and 2/6 had previous heart failure, but 2/6 had no previous cardiac events. The three patients with previous heart failure had median plasma hsTnT levels of 21.1 ng/L (17.6 to 85.4 ng/L) and median NT-pro BNP levels of 455 ng/L (24 to 775 ng/L). The eight patients with previous atrial fibrillation episodes had median hsTnT plasma levels of 11.9 ng/L (3.8 to 46 ng/L) and median NT-proBNP plasma levels of 368 ng/L (10 to 775 ng/L).

## 4. Discussion

This descriptive study evaluated the serum biomarkers hsTnT and NT-proBNP in a large cohort of Friedreich’s ataxia adult patients. While the prevalence of elevated troponin is 1% in the general population, 34% of FA patients had elevated plasma hsTnT values [18]. This high prevalence is similar to previously reported abnormal elevated plasma troponin levels in 47% of younger patients with Friedreich ataxia [9]. We found increased plasma hsTnT levels in patients with FA, similar to those with hypertrophic cardiomyopathy, but lower than those with acute coronary events [19,20,21]. Levels were also within the same range (<50 ng/L) in more severe stages of FA cardiomyopathy, as reported by Weidmann et al. [5]. Baseline cardiac characteristics, such as ECG and troponin levels, should be available for each FA patient as “their reference value”; they could be useful to manage further clinical acute events, such as chest pain. As in HCM, where troponin is positively correlated with maximum LV wall thickness, we found that increased septal wall thickness was associated with elevated plasma hsTnT levels [19,21]. However, hsTnT levels were not increased in all FA patients with hypertrophy, the biomarker could not replace the assessment of cardiac hypertrophy by echocardiography yet. Furthermore, we did not find a correlation between hsTnT and left ventricular systolic function as it has been previously reported in 32 FA patients [5].

In FA characterized by mitochondrial dysfunction, elevated hsTnT concentrations could reliably reflect ongoing subclinical myocyte injury, independently of coronary disease, but the precise mechanism of myocyte injury and release of hsTnT remain unresolved. One of the hypotheses is related to ferric/ferrous ion mediated generation of toxic oxygen species. Multifocal death and dropout of cardiomyocytes also leads to cardiac fibrosis [22,23]. Weideman et al. reported that hsTnT levels were raised in FA subjects with late gadolinium enhancement in cardiac MRI [5]. Fibrosis is likely to be an important feature of the cardiac disease and may play a role in prognosis. As it has been shown in hypertrophic cardiomyopathy, plasma hsTnT levels could predict cardiovascular events in FA [19].

We measured for the first time NT-proBNP plasma levels in a large cohort of FA patients. We found that only a small proportion of ambulatory FA patients (14%) had elevated NT-proBNP plasma levels. High plasma NT-proBNP levels were associated with parameters of left ventricular diastolic function and probably reflecting elevated left ventricular filling pressure in these patients. Indeed, the highest plasma levels were observed in patients with previous heart failure or atrial fibrillation. Whereas NT-proBNP plasma levels appeared to be a marker of early cardiac involvement in Fabry’s disease, in Friedreich ataxia they seemed to increase in more severe patients with previous cardiac events, likely signifying the evolution of the cardiac disease [11,24].

In FA, plasma levels of hsTnT and NT-proBNP were not associated and provided different types of information. Thus, in the study six patients had significant elevated levels of both with plasma hsTnT levels ≥14 ng/L and plasma NT-proBNP levels ≥ 125 ng/L. Three of these six patients had previous heart failure and two had had previous atrial fibrillation, whereas two patients of six had had no previous cardiac events. These two patients with high plasma levels of both biomarkers above the limits and no cardiac history may be at higher risk for cardiac complications.

The neurologic disease assessed by the Scale for the Assessment and Rating Ataxia (SARA) score [13] or the genetic abnormality were not associated with these cardiac biomarkers.

## 5. Limits of the Study

Our data concerned mainly adult FA patients recruited in a single center; thus, these findings could be generalized only to adult patients with FA. It would be of great interest to compare our adult population to pediatric subjects with more severe cardiac disease.

Additional limitations include the small patient sample size due to the relative rarity of this disease. Further larger collaborative studies are needed to confirm the range of plasma levels of NT-proBNP and hsTnT in FA patients and to define the place of these biomarkers in the management of FA patients.

## 6. Conclusions

Cardiac biomarkers such as NT-proBNP and hsTnT plasma levels are easily available and could be included in the routine cardiac evaluation of FA patients, in order to have individual reference values for the disease. In FA, hsTnT could be proposed as a marker of myocardial injury and cardiac involvement, and NT-proBNP would remain a marker of left ventricular filling pressure. Serial measurements are needed to characterize the temporal course of the two biomarkers and their relations with the evolution of echocardiographic parameters and the underlying cardiac disease. Further longitudinal studies are required to assess the prognostic value of these biomarkers in Friedreich ataxia.

## Figures and Tables

**Figure 1 jcm-09-01630-f001:**
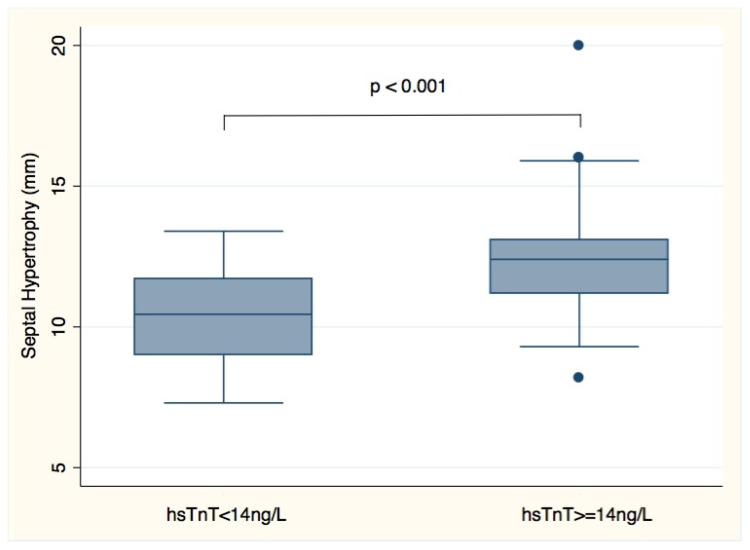
Plasma hsTnT levels and septal hypertrophy (septal wall thickness) in 70 Friedreich Ataxia patients.

**Table 1 jcm-09-01630-t001:** Clinical, genetic, electrocardiographic, and echocardiographic characteristics of 85 patients with Friedreich ataxia.

Parameters	Patients (*n* = 85) Mean ± SD
Age (years)	39 ± 12
Male sex *n* (%)	44 (51)
Body surface area (m^2^)	1.8 ± 1.7
GAA1 pb	480 ± 300
GAA2 pb (*n* = 83)	750 ± 300
Age at onset (years) (*n* = 83)	17 ± 10
Age at becoming wheelchair bound (years) (*n* = 53),	26 ± 9
Systolic blood pressure (mm Hg)	117 ± 12
Diastolic blood pressure (mm Hg)	75 ± 13
Heart rate (beats/min)	77 ± 13
Septal wall thickness (mm)	11 ± 2
Posterior wall thickness (mm)	10 ± 2
Relative wall thickness (RWT) > 0.42, *n* (%)	60 (71%)
Left ventricular diastolic diameter (mm)	45 ± 5
Left ventricular systolic diameter (mm) (*n* = 82)	27 ± 6
Left ventricular ejection fraction (%)	64 ± 6
Left ventricular mass index (g/m^2^)	98 ± 24
Trans mitral E wave (cm/s) (*n* = 84)	70 ± 10
Trans mitral A wave (cm/s) (*n* = 81),	60 ± 20
Trans mitral E/A ratio (*n* = 81)	1.3 ± 0.6
Lateral E/E’ ratio (*n* = 84)	6.2 ± 2.2
Left atrial area (cm^2^) (*n* = 83)	17 ± 5

GAA1 number of guanine-adenine-adenine (GAA). repeats in the shorter allele of the frataxin (FXN) gene; GAA2 number of GAA repeats in the larger allele of the FXN gene; E’ peak E’ velocity measured at the lateral wall (tissue Doppler imaging).

**Table 2 jcm-09-01630-t002:** Parameters associated with elevated plasma high-sensitivity troponin (hsTnT) levels in 70 patients with Friedreich ataxia. NT-proBNP: N-terminal pro-brain natriuretic peptide; LV: left ventricle.

Parameters	Univariate Analysis		Multivariate Analysis	
	OR (95%IC)	*p*	OR (95%IC)	*p*
Age	0.96 (0.92–1.00)	0.05 *	-	NS
Male sex	0.88 (0.33–2.34)	0.79		
Body surface area	0.40 (0.02–7.08)	0.36		
GAA1	1.47 (0.67–3.18)	0.29		
GAA2	1.32 (0.69–2.55)	0.72		
Heart rate	1.02 (0.98–1.06)	0.31		
NT-proBNP plasma level ≥ 125 ng/L	2.34 (0.63–8.61)	0.20		
Septal wall thickness	1.79 (1.27–2.53)	<0.001 *	1.79 (1.27–2.53)	<0.001
Posterior wall thickness	1.38 (1.01–1.88)	0.04 *	-	NS
Relative wall thickness (RWT) > 0.42	2.64 (0.83–8.34)	0.1	-	
LV diastolic diameter	0.99 (0.9–1.09)	0.66		
LV systolic diameter	1.06 (0.97–1.15)	0.19		
LVejection fraction	0.91 (0.84–1.00)	0.17		
LV mass index	1.04 (1.01–1.08)	<0.001 *	-	NS
Trans mitral E wave	0.1 (0.001–6.06)	0.59		
Trans mitral A wave	0.32 (0.01–6.84)	0.53		
E/A ratio	1.27 (0.58–2.77)	0.6		
Lateral E/E’ ratio	1.1 (0.86–1.51)	0.5		

* variables included in the multivariate analysis. NS: non significant.

**Table 3 jcm-09-01630-t003:** Parameters associated with elevated plasma NT-proBNP levels in 85 patients with Friedreich ataxia.

Parameters	Univariate Analysis		Multivariate Analysis	
	OR (95%IC)	*p*	OR (95%IC)	*p*
Age	0.99 (0.94–1.04)	0.72		
Male sex	2.42 (0.67–8.76)	0.17		
Body surface area	1.34 (0.03–48.7)	0.87		
GAA1	4.10 (1.25–13.38]	0.02 *	-	NS
GAA2	1.95 (0.84–4.53)	0.12		
Heart rate	1.00 (0.96–1.05)	0.79		
Septal wall thickness	1.41 (1.06–1.89)	0.02 *	-	NS
Posterior wall thickness	1.14(0.78–1.66)	0.49		
Relative wall thickness (RWT) > 0.42	5.38 (0.65–44.19)	0.11		
LV diastolic diameter	0.99 (0.88–1.12)	0.95		
LV systolic diameter	1.05 (0.94–1.17)	0.34		
LV ejection fraction	0.88 (0.80–0.98)	0.02 *	-	NS
LV mass index	1.02 (0.99–1.04)	0.11		
Trans mitral E wave	0.20 (0.01–30)	0.53		
Trans mitral A wave	0.01 (0.00–0.37)	0.02 *	-	NS
E/A ratio	3.46 (1.30–9.19)	0.01 *	3.48 (1.28–9.48)	0.01
Lateral E/E’ ratio	1.47 (1.06–2.02)	0.02 *	1.64 (1.13–2.40)	0.01
Left atrial area	1.20 (1.05–1.36)	0.006 *	-	NS

* variables included in the multivariate analysis. NS: non significant.

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
