# Peer review of "Significance of NT-proBNP and High-Sensitivity Troponin in Friedreich Ataxia"

_jcm, 2020, doi:10.3390/jcm9061630_

Round 1
Reviewer 1 Report
Legrand et al. prospectively performed measurements of hsTnT and NT-proBNP as biomarkers in 85 Friedreich Ataxia (FA) patients in order to determine the utility of using these biomarkers to assess the progression of cardiomyopathy in FA patients. Both elevated hsTnT and NT-proBNP correlated with various markers of hmodynamic compromise in the cohort of patients. The data support the authors' conclusion that these biomarkers can be used to identify FA patients that should undergo a closer cardiac surveillance.
The main deficiency of the study is the lack of correlation with what is known about the underlying pathology of the cardiomyopathy in FA. This has been well characterized in the literature. For example, see:
Koeppen AH. Friedreich's ataxia: pathology, pathogenesis, and molecular genetics. J Neurol Sci. 2011;303(1-2):1‐12. doi:10.1016/j.jns.2011.01.010.
Patients may develop cardiac hypertrophy simulating genetic hypertrophic cardiomyopathy. Additionally, the cardiomyopathy progresses as a result of cardiomyocyte injury, probably related to ferric/ferrous ion mediated generation of toxic oxygen species. Multifocal death and dropout of cardiomyocytes leads to progressive cardiac fibrosis.
This provides a rational explanation for the observed TnT leak and elevated serum TnT as well as increased expression and release of NT-proBNP.
A minor correction, but an important point, is that GAA should be defined as a guanine-adenine-adenine trinucleotide.
Also, what is the SARA score? Give a reference.
Author Response
The main deficiency of the study is the lack of correlation with what is known about the underlying pathology of the cardiomyopathy in FA. This has been well characterized in the literature. For example, see:
Koeppen AH. Friedreich's ataxia: pathology, pathogenesis, and molecular genetics. J Neurol Sci. 2011;303(1-2):1‐12. doi:10.1016/j.jns.2011.01.010.
Patients may develop cardiac hypertrophy simulating genetic hypertrophic cardiomyopathy. Additionally, the cardiomyopathy progresses as a result of cardiomyocyte injury, probably related to ferric/ferrous ion mediated generation of toxic oxygen species. Multifocal death and dropout of cardiomyocytes leads to progressive cardiac fibrosis. This provides a rational explanation for the observed TnT leak and elevated serum TnT as well as increased expression and release of NT-proBNP.
Thank you we modified the discussion. Line 274 to line 276: One of the hypothesis is probably related to ferric/ferrous ion mediated generation of toxic oxygen species. Multifocal death and dropout of cardiomyocytes also leads to progressive cardiac fibrosis:
We also add two references in the bibliography
A minor correction, but an important point, is that GAA should be defined as a guanine-adenine-adenine trinucleotide.
We corrected it in the introduction: line70 :FA is in most cases caused by a homozygous guanine-adenine-adenine (GAA) trinucleotide expansion in the frataxin (FXN) gene
Also, what is the SARA score? Give a reference
The SARA (Scale for the Assessment and Rating of Ataxia), is a tool for assessing ataxia with eight categories with accumulative score ranging from 0 (no ataxia) to 40 (most severe ataxia). When completing the outcome measure each category is assessed and scored accordingly.
We explained it in the material and method part line 112-115 supported by reference 13
Reviewer 2 Report
A very well done study indicating on the implication of TnT and BNP testing in FA as one example of metabolic CMP.
The point is nearly all had normal or near-normal echo parameters.
What happens in patients with advanced disease? Do we have at least some data on follow up, repeat measures and disease progression?
As FA causes cardiac manifestation reminiscent of HCM, please provide some overview of TnT and BNP levels in early HCM and their implication.
Author Response
The point is nearly all had normal or near-normal echo parameters.
Thank you for your comment: in fact mean echo parameters values are near normal. In Friedreich ataxia there is a concentric mild hypertrophy with increased wall thicknesses associated with reduced left ventricular end diastolic diameter, therefore LV mass can remain in the normal range. Most patients have increased related wall thickness (RWT) and therefore myocardial involvement. Reduced Left ventricular ejection fraction may occurred over time and is of worse prognosis.
What happens in patients with advanced disease? Do we have at least some data on follow up, repeat measures and disease progression?
In this study 2 patients had systolic dysfunction and previous heart failure, one patient is alive (70 years), the other young patient is dead last year at the age of 30 (sudden death)
Heart is the main cause of death for FA patients, heart failure and supraventricular arrhythmia occurred respectively in 8.3% and 16.5% in a previous larger study.(6, 7).
Concerning the present study we have no data concerning the repetition of cardiac biomarkers measurements. We agree that it would be interesting to study the temporal course of Hstroponin and Nt-proBNP plasma levels and their relations with the evolution of the underlying cardiac disease.As FA causes cardiac manifestation reminiscent of HCM, please provide some overview of TnT and BNP levels in early HCM and their implication.
Literature is poor concerning TnT and natriuretic peptide in genetic HCM:
Kubo et al (19) showed a positive correlation between maximal LV wall thickness and HsTnT plasma levels. We mentioned it in the discussion line 267.
Jenab et al showed that 43 % of HCM patients had increased levels of hs-TnT ≥ 14ng/L
Miyaji et al in Intern Med 2016 showed that Left ventricular mass assessed with CMR analysis was correlated with bnp level.
Both cardio biomarkers are associated with cardiovascular events in HCM.